

# Visualization of Spin Dynamics in Coupled Nuclear Multispin Systems

Jingyan Xu[1,2], Dmitry Budker[1,2,3], and Danila A. Barskiy[1,2]

[1]Institut für Physik, Johannes Gutenberg Universität Mainz, 55128 Mainz, Germany
[2]Helmholtz Institut Mainz, 55128 Mainz, Germany; GSI Helmholtzzentrum für Schwerionenforschung, Darmstadt, Germany
[3]University of California at Berkeley, California 94720-7300, USA

**Correspondence:** Danila Barskiy (dbarskiy@uni-mainz.de)

**Abstract.** Since the dawn of quantum mechanics ways to visualize spins and their interactions attracted attention of researchers and philosophers of science. Angular momentum probability (AMP) surfaces are known for visualizing density matrices; a plotted surface represents probability of finding maximum projection of angular momentum along any direction in space. While AMP surfaces visually reveal symmetries of density matrices which directly translate to measured properties, they focus
solely on one total-angular-momentum manifold, neglecting interaction between different manifolds often encountered in the NMR of multispin systems. In this work, we extend applicability of this visualization method and introduce its extension, angular momentum coherence (AMC) surface approach. Three examples of nuclear spin dynamics in two-spin systems are presented and visualized using the AMP/AMC surfaces: (I) spin-lock induced crossing (SLIC) sequence with initial state being the singlet state between two spins; (II) interplay between chemical exchange and spin dynamics during the high-field
signal amplification by reversible exchange (SABRE) experiment; (III) zero- to ultralow-field (ZULF) NMR experiment in the presence of magnetic field applied perpendicular to the sensitive axis of the detector. The presented visualization technique extends applicability of AMP surfaces to coupled multispin systems and will facilitate intuitive understanding of spin dynamics during complex NMR experiments as exemplified here by the considered cases. The temporal sequences ("the movies") of such surfaces show phenomena like interconversion of polarization moments (Auzinsh et al., 2010) and, as particularly relevant in
NMR and demonstrated here, polarization transfer between different spins. Such effects are difficult to grasp by looking at (time-dependent) numerical values of density-matrix elements.

## 1  Introduction

The evolution of spins in nuclear magnetic resonance (NMR) experiments can be highly complex and non-intuitive, thus, approaches to visualize spin dynamics of nuclear multispin systems is sought for both communication and research purposes.
Most NMR textbooks visualize the motion of a single spin-1/2 (or an ensemble of spins) using the Bloch vector (e.g., Bloch, 1946; Feynman et al., 1957; Levitt, 2013). In more complex cases such as coupled spin systems, visualization is significantly less straightforward, and the discussion is often assisted by drawing energy level diagrams. These diagrams do not provide dynamic information but can be useful for representing populations of various spin states and coherences between them (e.g., Sørensen et al., 1984; Messiah, 1999; Levitt, 2013; Barskiy et al., 2019).





Recently, Garon et al. introduced an approach of so-called "Spin Drops" (DROPS = discrete representation of operators for spin systems) for visualizing spin operators in NMR experiments (Garon et al., 2015; Leiner et al., 2017; Leiner and Glaser, 2018). This visualization approach is based on plotting 3-dimensional colorful shapes (drops) that correspond to linear combinations of spherical harmonics. Spin drops can represent interactions in Hamiltonians and in propagators, as well as states of the density matrix; thus, spin drops can be used to describe the dynamics during each stage of NMR experiment

(Garon et al., 2015; Leiner et al., 2017; Leiner and Glaser, 2018). While the approach conveniently reflects the dynamics and symmetry of individual spins (which is useful in high-field NMR experiments where spins can be addressed individually), it is challenging to extract information from the drops representing systems of equivalent (or nearly-equivalent) spins. The rotational symmetry of the total nuclear spin is not broken, for example, in NMR experiments utilizing parahydrogen ($p\mathrm{H}_2$) and in zero- to ultralow-field (ZULF) NMR experiments. To address these limitations, we introduce a generalized approach

based on the angular momentum probability (AMP) surfaces (Auzinsh, 1997; Rochester and Budker, 2001; Auzinsh et al., 2010).

The angular momentum probability surface approach was first introduced by Rochester et al. for visualizing the angular momentum state of atoms (Rochester and Budker, 2001; Auzinsh et al., 2010). In this approach, the distance of each point of the visualized surface from the origin of the coordinate system is equal to the probability of finding the maximal possible

projection along the radial direction. However, AMP surfaces do not represent coherences connecting states with different total angular momentum (including states of the same total angular momentum quantum number $F$ but belonging to different manifolds) which play a crucial role in NMR of multispin systems. In this work we demonstrate that coherences can also be visualized by plotting specific observables along different directions, thus, called hereafter AMC (angular momentum coherence) surfaces. Together with AMP surfaces, the approach presented in this work constitutes convenient means for vizualizing

complex dynamics in multispin systems exemplified here for pairs of nuclear spins 1/2.

## 2 Results

Our generalized visualization approach is pictorially shown in Fig.1. We first write down the density matrix (DM) in the total angular momentum basis and decompose it into blocks according the total angular momentum quantum number $F$ (Fig. 1A).

In order to plot the AMP surface for the block $(F_\mathrm{k}, F_\mathrm{k})$ (where $k$ is used to distinguish different $F$ blocks), the probability

of finding the maximal projection state in the block is assigned to the distance $r$ from the origin (Fig. 2) along the direction $\hat{\mathbf{n}}$ such that

$$r(\theta, \phi) = {}_{\hat{\mathbf{n}}}\langle F_\mathrm{k}, F_\mathrm{k} | \hat{\rho} | F_\mathrm{k}, F_\mathrm{k} \rangle_{\hat{\mathbf{n}}} = \mathrm{Tr}(\hat{\rho} | F_\mathrm{k}, F_\mathrm{k} \rangle_{\hat{\mathbf{n}}} \, {}_{\hat{\mathbf{n}}}\langle F_\mathrm{k}, F_\mathrm{k} |), \tag{1}$$

where $|F_\mathrm{k}, F_\mathrm{k}\rangle_{\hat{\mathbf{n}}}$ is the state with maximal projection along measurement direction $\hat{\mathbf{n}}$ (see Eq. (C4)).

When it comes to off-diagonal blocks $(F_\mathrm{l}, F_\mathrm{k})$, one can consider a zero-quantum coherence ($\Delta m = 0$) of a specific form

defined along the measurement direction $\hat{\mathbf{n}}$,

$$\left( \widehat{\mathrm{ZQ}}_{\varphi, m_\mathrm{F}}^{(F_\mathrm{l}, F_\mathrm{k})} \right)_{\hat{\mathbf{n}}} = e^{i\varphi} | F_\mathrm{l}, m_\mathrm{F} \rangle_{\hat{\mathbf{n}}} \, {}_{\hat{\mathbf{n}}}\langle F_\mathrm{k}, m_\mathrm{F} | + e^{-i\varphi} | F_\mathrm{k}, m_\mathrm{F} \rangle_{\hat{\mathbf{n}}} \, {}_{\hat{\mathbf{n}}}\langle F_\mathrm{l}, m_\mathrm{F} |, \tag{2}$$



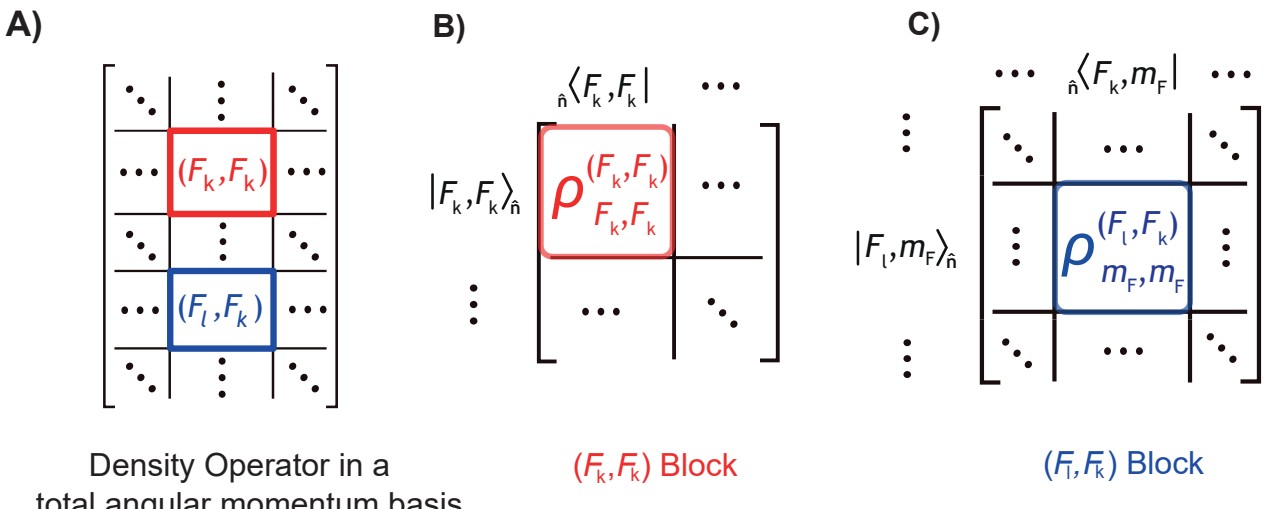

**Density Operator in a total angular momentum basis**

$(F_k, F_k)$ **Block**

$(F_l, F_k)$ **Block**

**Figure 1.** Visualization algorithm for a density matrix (DM): (A) Write down a DM in the total angular momentum basis and decompose it into blocks according to $F$; (B) Write down matrix elements of the diagonal block $(F_k, F_k)$ in the basis with the quantization axis set along the direction $\hat{\mathbf{n}}$, the top left element corresponds to the probability of finding the maximal projection along this direction; (C) Write down matrix elements of the blocks $(F_l, F_k)$ and $(F_k, F_l)$ in the basis with the quantization axis set along the direction $\hat{\mathbf{n}}$, a combination of elements with $m_l = m_k$ are used for the visualization.

where operators are defined up to a phase $\varphi$ and added with their conjugate transpose pairs to be Hermitian, i.e., to correspond to the proper observable quantities; $|F_l, m_F\rangle_{\hat{\mathbf{n}}}$ is the state with a projection $m_F$ along the measurement direction $\hat{\mathbf{n}}$.

For implementing visualization of the AMC surface, a projection quantum number $m_F$ is chosen within the range $|m_F| \leq$
$\min(F_l, F_k)$. Given a fixed $m_F$, one may notice that the defined operators $(\widehat{ZQ}_{\varphi, m_F}^{(F_l, F_k)})_{\hat{\mathbf{n}}}$ vary only by one number, $\varphi$, and form a two-dimensional real operator space. Therefore, it is enough to consider only two values of $\varphi$ to fully represent spin dynamics of the coherences between blocks $(F_k, F_k)$ and $(F_l, F_l)$; in this work we pick $\varphi$ equal to $0$ and $\pi/2$.

For visualizing an AMC surface, we assume measurements according to the defined observables along various directions in a manner similar to the AMP surface:

$$r(\theta, \phi) = \mathrm{Tr}\left[\hat{\rho}\left(\widehat{ZQ}_{\varphi, m_F}^{(F_l, F_k)}\right)_{\hat{\mathbf{n}}}\right],$$
                  (3)

The expectation value for the operator $(\widehat{ZQ}_{\varphi, m_F}^{(F_l, F_k)})_{\hat{\mathbf{n}}}$ can be negative, thus, an additional "degree of freedom", color, is introduced. As a convention, in this work we set the color to be red if the expectation value is positive and blue if the expectation value is negative. Since we initially chose observable operators to be Hermitian, additional colors are not necessary. However, if the observable operators are not set to be Hermitian, a color map is necessary to express $(0, 2\pi)$-phase dependence of the
computed expectation value which is a complex number.



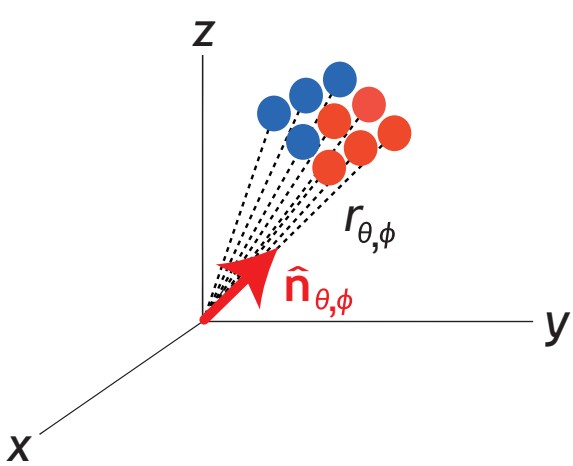

**Figure 2.** A single surface point along the direction $\hat{\mathbf{n}}$ (equivalently denoted by $(\theta, \phi)$) and $r$ is the distance from the surface to the origin.

We note that with our generalized visualization approach, one can see the symmetries of the quantum states, for instance, the $q$-fold rotational symmetries that arise when only particular polarization moments are present. For example, when all elements are zero except those for which $m_l - m_k = qN$, where $N$ is a integer, then the AMP/AMC surfaces possess a $q$-fold symmetry, see Appendix D. (Auzinsh et al., 2010, p. 100)

As an example, consider a pair of spin-1/2 nuclei. There are in total four blocks. For the two diagonal blocks ($F{=}1, F{=}1$) and ($F{=}0, F{=}0$), one can draw AMP surfaces by considering the probabilities of finding the states $|1,1\rangle_{\hat{\mathbf{n}}}$ and $|0,0\rangle$, respectively. Four examples of different AMP surfaces ($F{=}1, F{=}1$) are presented in Fig. 3. The first example shown in Fig. 3A is a polarized state that exemplifies magnetic orientation, i.e., a preferred direction of magnetization in space. The second example Fig. 3B shows an anti-phase spin order exemplifying alignment (Auzinsh et al., 2010). Such spin order is often an

initial state in parahydrogen-based hyperpolarization experiments (Bowers and Weitekamp, 1987; Natterer and Bargon, 1997). The spin orders shown in Fig. 3C and Fig. 3D are often encountered in various experiments involving two-spin systems (see the discussion below). The usefulness of our visualization approach becomes apparent when one can spot the presence of different symmetries (or lack of thereof) by simply looking at the calculated surface without analyzing the explicit structure of the density matrices.

The two remaining off-diagonal blocks are mapped into AMC surfaces, and the only quantum number $m_F{=}0$ is permissible due to the requirement $|m_F| \leq \min(1, 0)$. Taking into consideration the discussed constraint on $\varphi$, one can show that

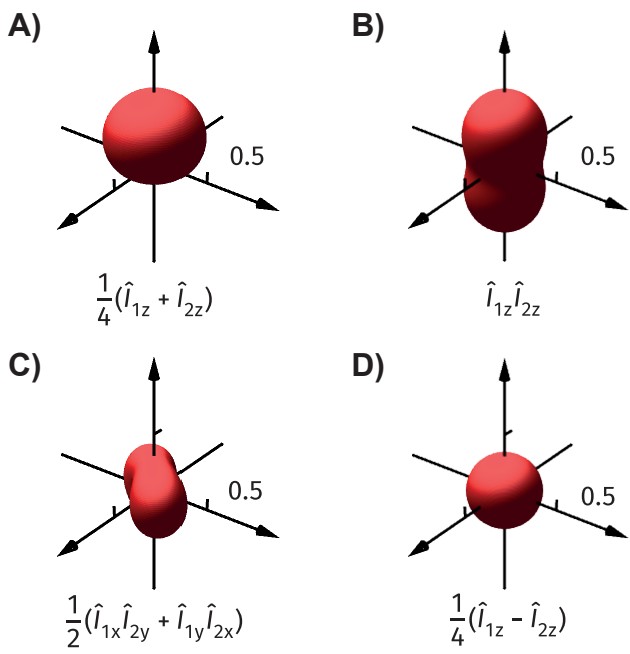

**Figure 3.** Visualizations using AMP surface of the block $(F=1, F=1)$ for the following density operators: (A) $\hat{\rho} = \frac{1}{4}\mathbb{1} + \frac{1}{4}(\hat{I}_{1z} + \hat{I}_{2z})$, representing a state orientated along $\hat{\mathbf{z}}$, (B) $\hat{\rho} = \frac{1}{4}\mathbb{1} + \hat{I}_{1z}\hat{I}_{2z}$, representing a state aligned along $\hat{\mathbf{z}}$, (C) $\hat{\rho} = \frac{1}{4}\mathbb{1} + \frac{1}{2}(\hat{I}_{1x}\hat{I}_{1y} + \hat{I}_{2y}\hat{I}_{1x})$, representing a state with a more complex alignment, (D) $\hat{\rho} = \frac{1}{4}\mathbb{1} + \frac{1}{4}(\hat{I}_{1z} - \hat{I}_{2z})$, representing a state that appears isotropic from the point of view of the total angular momentum with $F = 1$ (the state is intrinsically anisotropic and the AMP surface cannot visualize it; AMC surface should be used instead).

$(\widehat{ZQ}_{0,0}^{(1,0)})_{\hat{\mathbf{z}}} = (\hat{I}_{1z} - \hat{I}_{2z})$ (i.e., magnetization difference) when $\varphi = 0$, and $(\widehat{ZQ}_{\frac{\pi}{2},0}^{(1,0)})_{\hat{\mathbf{z}}} = (\hat{I}_{1y}\hat{I}_{2x} - \hat{I}_{1x}\hat{I}_{2y})$ (the so-called zero-quantum coherence out of phase (Pravdivtsev et al., 2017)) when $\varphi = \pi/2$. For visualization of AMC surfaces, we then plot the expectation values $\langle(\widehat{ZQ}_{0,0}^{(1,0)})_{\hat{\mathbf{n}}}\rangle$ and $\langle(\widehat{ZQ}_{\frac{\pi}{2},0}^{(1,0)})_{\hat{\mathbf{n}}}\rangle$ according to the equation (3).

We note here the AMP/AMC surface approach is not limited to coupled spin-1/2 nuclear spin systems. For example, in a system of coupled spin-1/2 and spin-1 nuclei, two AMP surfaces for the diagonal blocks, $(F=\frac{3}{2}, F=\frac{3}{2})$ and $(F=\frac{1}{2}, F=\frac{1}{2})$, and two AMC surfaces for the two off-diagonal blocks, $(F=\frac{3}{2}, F=\frac{1}{2})$ and $(F=\frac{1}{2}, F=\frac{3}{2})$, would be necessary to visualize spin dynamics. AMP/AMC surfaces could in principle be used for describing dynamics of interacting electron spins during experiments such as DNP (Dynamic Nuclear Polarization), CIDNP (chemically-induced dynamic nuclear polarization) and

general electron paramagnetic resonance (EPR) experiments. However, since the examples in our paper are focused solely on NMR, in the following we limit our discussion to nuclear spin systems only.



## 3 Discussion

### 3.1 Spin-Lock Induced Crossing

Figure 4 shows a typical spin-lock induced crossing (SLIC) experiment where a continuous radio frequency (RF) field is applied
to a pair of chemically-inequivalent nuclear spins at high field, initially prepared in the singlet spin state. The target spins
undergo singlet-triplet conversion if the spins are strongly coupled to each other and the amplitude of the RF field matches
the $J$-coupling value between them (DeVience et al., 2013). The parameters used for the simulation and visualization are
shown in Fig. 4A where the molecule is specified by $J$-coupling strength between the two spins and $\Delta$, the difference between
their Larmor frequencies, expressed in hertz (Levitt, 2013). Under the above conditions, the singlet and triplet populations
interconvert with a period $T = \frac{\sqrt{2}}{\Delta}$.

We simulate the SLIC experiment by considering only the coherent processes (no relaxation is included). One could process
the simulation results by plotting the population of several chosen operators as a function of time, and the derived population
plot is shown in Fig. 4B. Note that all these operators are defined in the frame rotating in sync with the RF-field. Figure 4B
illustrates that the singlet spin order is converted – upon application of a SLIC pulse – into the magnetization opposite to the
direction of the RF-field amplitude in the rotating frame. Note here the conversion of the singlet spin order into magnetization
parallel to the direction of the RF-field is symmetry-allowed but energy-forbidden, which introduces small but fast oscillations
into the AMP/AMC surfaces.

### 3.2 Signal Amplification By Reversible Exchange

As a second example, we visualize dynamics of the nuclear spin states of dissolved hydrogen during signal amplification by
reversible exchange (SABRE) experiments (Adams et al., 2009) at high magnetic field (Barskiy et al., 2014). The interplay
between chemical exchange and coherent spin dynamics is known to induce singlet-triplet mixing (Kiryutin et al., 2017; Knecht
et al., 2019; Markelov et al., 2021). For the two protons in free hydrogen (i.e., molecular hydrogen gas dissolved in solution), the
singlet triplet-mixing is symmetry-forbidden. However, as soon as the hydrogen molecule binds transiently to a SABRE catalyst
and the two protons occupy non-equivalent positions such that chemical symmetry is broken, the difference in their Larmor
frequencies, $\Delta = (\delta_1 - \delta_2)\gamma B_0$, gives rise to singlet-triplet mixing. Such mixing is selective and only $|S\rangle \to |T_0\rangle$ transitions
occur, while the other transitions $|S\rangle \to |T_\pm\rangle$ are forbidden at high field. In the following, we also consider relaxation via
locally correlated noise fields to account for the equilibration of triplet sublevels (Ivanov et al., 2008).

We apply a model recently introduced to simulate spin dynamics in SARBE experiments. All the parameters used for the
simulation are given in Fig. 5A where the chemical rate constants $k_a$ and $k_d$ describe the rates of association and dissociation
of $H_2$, respectively, and the longitudinal relaxation time of the two protons is given by $T_1^f$ when they are free in solution and $T_1^b$
when they are bound to the SABRE catalyst, respectively. The simulated state populations are shown in Fig. 5B. First, after start
of the exchange process, chemical asymmetry induces the $|S\rangle \to |T_0\rangle$ transition and reduces the population difference between
the two states. As the $|T_0\rangle$ state is populated, relaxation between triplet sublevels becomes more noticeable and the states $|T_\pm\rangle$

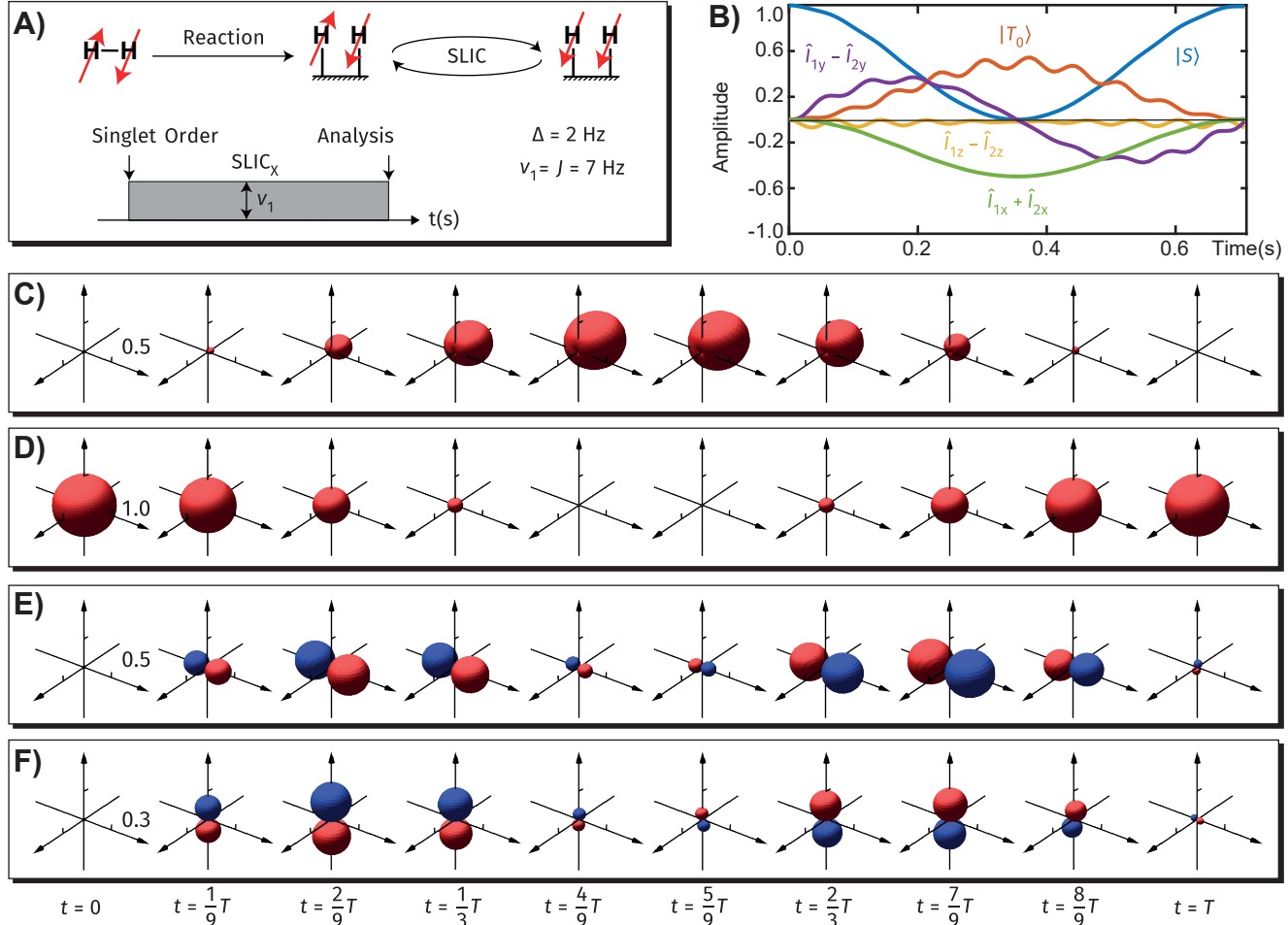

**Figure 4.** Visualization of spin dynamics in proton pairs during high-field SLIC experiments over a period $T = \sqrt{2}/\Delta$. (A) Scheme of the SLIC experiment: an RF-field of the amplitude $B_1$ equal to $J$-coupling between the spins is applied along rotating-frame $x$-axis at the average proton resonance frequency. (B) Evolution of various spin orders during the SLIC experiment. (C-F) Visualization of the evolution under SLIC experiment using (C) AMP surface of the maximum projection state $|11\rangle_{\hat{n}}$, (D) AMP surface of the singlet population, (E) AMC surface $\langle(\widehat{ZQ}_{0,0}^{(1,0)})_{\hat{n}}\rangle$, (F) AMC surface $\langle(\widehat{ZQ}_{\frac{\pi}{2},0}^{(1,0)})_{\hat{n}}\rangle$. Note different scaling in (C)-(F).

also start to be populated. Since the local noise fields are not perfectly correlated, the populations are finally equalized as shown in Fig. 5B.

The evolving distribution of population over various states is seen in Fig. 5C-D. In Fig. 5C, the AMP surface for the triplet block first grows into an oblate spheroid which indicates that initially only the state $|T_0\rangle$ state gets overpopulated. Later, the AMP surface evolves into a sphere indicating that all three triplet states are equally populated as a result of relaxation. Figure 5D shows the changing singlet population. Lastly, Fig. 5E, which measures the out-of-phase coherence $\langle(\widehat{ZQ}_{\frac{\pi}{2},0}^{(1,0)})_{\hat{n}}\rangle$,



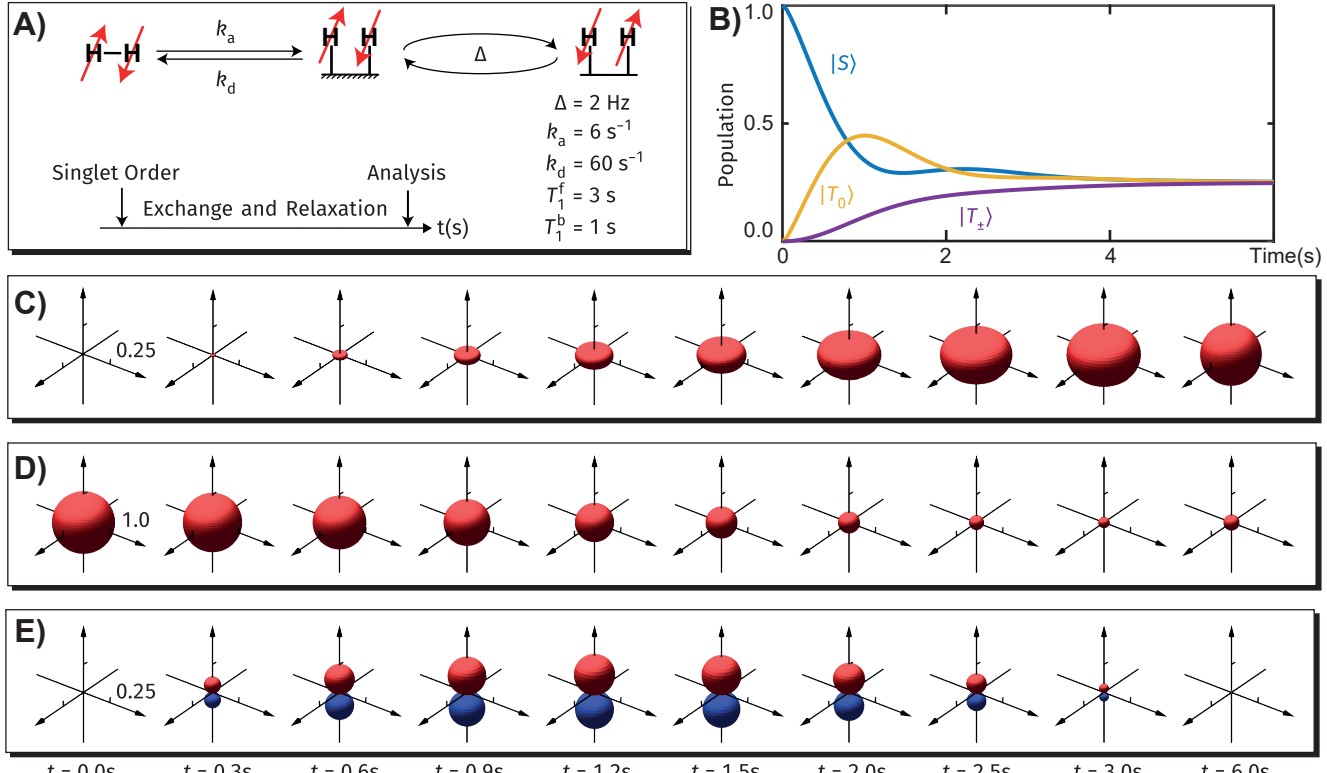

**Figure 5.** Visualizations of spin dynamics in proton pairs of free molecular hydrogen in a SABRE experiment. (A) Scheme of the SABRE experiment: molecular hydrogen initiated in the singlet state undergoes reversible chemical coordination which transiently breaks chemical equivalence between the two spins; NMR, exchange, and relaxation parameters are shown in the insert. (B) Evolution of the spin state populations during the SABRE experiment. (C-F) Visualization of the evolution under SABRE experiment using (C) AMP surface of the state with maximal projection $|11\rangle_{\hat{\mathbf{n}}}$, (D) AMP surface of the singlet population, and (E) AMC surface $\langle(\widehat{ZQ}^{(1,0)}_{\frac{\pi}{2},0})_{\hat{\mathbf{n}}}\rangle$. Note different scaling in (C) and (D).

135    reveals extra information not covered by Fig. 5B, i.e., the singlet-triplet coherence, $\hat{I}_{1y}\hat{I}_{2x} - \hat{I}_{1x}\hat{I}_{2y}$, is transiently formed during the experiment. Since the $J$-coupling between the two protons is ignored, there is no in-phase coherence (i.e., magnetization differences between the two protons along any direction) and the related surface plot is therefore not included here.

## 3.3   Zero- to Ultralow-Field Nuclear Magnetic Resonance

Lastly, we show how one could use the presented visualization approach to better understand spin dynamics in zero- to ultralow-

140   field (ZULF) NMR experiments (Ledbetter et al., 2011). As an example, we take $^{13}$C-labeled formic acid with $^1$H and $^{13}$C spins initially polarized along the $z$ axis (direction of a magnetometer's sensitive axis). Before acquisition, $^{13}$C spins are selectively inverted (Sjolander et al., 2016; Jiang et al., 2018) to increase the signal, and during the acquisition a weak bias magnetic field,



**Figure 6.** Visualizations of spin dynamics in an AX system ($^1$H-$^{13}$C nuclear pair) during zero- to ultralow-field nuclear magnetic resonance (ZULF) experiment. (A) Scheme of an exemplary ZULF NMR experiment in which perpendicular field of 2.64 mG is applied during the detection. (B) Simulation of the corresponding ZULF NMR spectrum. (C)-(F) AMP and AMC surfaces representing spin evolution in the ZULF experiment. (C) The state with maximal projection $|11\rangle_{\hat{\mathbf{n}}}$ over a time scale of $T_1 = 2/(\nu_{1H} + \nu_{13C})$. (D) In-phase zero quantum coherence $\langle(\widehat{ZQ}_{0,0}^{(1,0)})_{\hat{\mathbf{n}}}\rangle$ shown over a time scale of $T_1$; (E) In-phase zero quantum coherence $\langle(\widehat{ZQ}_{0,0}^{(1,0)})_{\hat{\mathbf{n}}}\rangle$ shown over a time scale of $T_2 = 1/J$. (F) Zero quantum coherence out-of-phase $\langle(\widehat{ZQ}_{\frac{\pi}{2},0}^{(1,0)})_{\hat{\mathbf{n}}}\rangle$ shown over a time scale of $T_2$.

$B_x$, perpendicular to the magnetometer's sensitive axis is applied (Fig. 6A). In such an experiment, nuclear spins evolve due to the heteronuclear $J$-coupling and the bias field, the ZULF NMR spectrum is collected by the measuring the magnetization

145 along the $z$ direction. A simulated NMR spectrum following the described procedure is shown in Fig. 6B. Note that only the



positive frequency region is plotted given only one measurement device. First, low-frequency peak is positioned at the average Larmor frequency between the proton and the carbon. In addition, a doublet with splitting equal to the sum of the Larmor frequencies of proton and carbon is centered at the *J*-coupling frequency.

Figure 6C shows the AMP surface by measuring the maximal projection state, the surface is rotating about the *x*-axis with a period of $T_1 = 2/(\nu_{^1H} + \nu_{^{13}C})$, where $\nu_{^1H}$ and $\nu_{^{13}C}$ are proton and carbon Larmor frequencies, respectively. One can instantly realize that this motion corresponds to the leftmost single peak in Fig 6B. Dynamics of the AMC surfaces shown in Fig 6D-F is more intricate. Their motion is a superimposed oscillation of shapes with a period of $T_2 = 1/J$ and a slow precession about the *x*-axis with the above-mentioned period of $T_1$. From the visualization, one may conclude that in-phase and out-of-phase zero quantum coherences give rise to the doublet shown in Fig 6B.

## 4 Conclusions

In this paper, we extend an approach for visualizing dynamics of quantum spin systems based on angular momentum probability (AMP) surfaces to visualizing coherences via angular momentum coherence (AMC) surfaces. AMP/AMC surfaces conveniently represent symmetries of density matrices and allow spotting their presence (orientation, alignment, etc.) or absence even when direct analysis of density matrices is not obvious. Three different experiments are used to demonstrate applicability of the novel visualization approach: (I) spin dynamics in the ensembles of pairs of spin-1/2 nuclei during the spin-lock induced crossing (SLIC) experiment; (II) physicochemical conversion of parahydrogen during signal amplification by reversible exchange (SABRE) experiment at high magnetic field; and (III) evolution of heteronuclear $^1$H-$^{13}$C spin pair during the zero- to ultralow-field (ZULF) NMR experiment. The presented AMP/AMC surface approach allows visualizing complex dynamics in multispin systems and may find applications for describing hyperpolarization experiments utilizing parahydrogen (PHIP and SABRE), experiments such as DNP (Dynamic Nuclear Polarization), CIDNP (chemically-induced dynamic nuclear polarization) and general NMR and EPR experiments.

*Code availability.* The software code for the graphics shown in this paper is available from the authors upon request.

*Data availability.* No data sets were used in this article.

*Video supplement.* Video representations of the figures are available in Supporting Information.



## Appendix A: Spherical Tensor Operator Basis

### A1 Definition of the spherical tensor basis operators

For each block $(F_l, F_k)$, we define a basis of tensor operators by

$$\hat{\mathbb{T}}_{\lambda\mu}^{(F_l, F_k)} = \sqrt{\frac{2\lambda+1}{2F_l+1}} \sum_{m_l, m_k} \langle F_k m_k \lambda\mu | F_l m_l \rangle | F_l, m_l \rangle \langle F_k, m_k |, \tag{A1}$$

with $|F_l - F_k| \leq \lambda \leq F_l + F_k$ and $-\lambda \leq \mu \leq \lambda$, and $m_k$, $m_l$, and $\mu$ being the projection quantum numbers of the Clebsch-Gordan coefficient $\langle F_k m_k \lambda\mu | F_l m_l \rangle$.

### A2 Proprieties of the spherical tensor basis operators

The defined operator basis satisfy the following proprieties:

A2.1. *Orthogonality*: $\mathrm{Tr}\left( \hat{\mathbb{T}}_{\lambda\mu}^{(F_l, F_k)} \left( \hat{\mathbb{T}}_{\lambda'\mu'}^{(F_{l'}, F_{k'})} \right)^\dagger \right) = \delta_{F_l F_{l'}} \delta_{F_k F_{k'}} \delta_{\lambda\lambda'} \delta_{\mu\mu'}$.

A2.2. *Completeness*: The operator basis is complete (in a sense that it can be used to represent any operator).

A2.3. *Sphericalness*: Given the rotation operator $\hat{R}(\alpha, \beta, \gamma)$, a tensor operator transforms under rotation as

$$\hat{R}(\alpha, \beta, \gamma) \hat{\mathbb{T}}_{\lambda,\mu}^{(F_l, F_k)} \hat{R}^\dagger(\alpha, \beta, \gamma) = \sum_{\mu'} \hat{\mathbb{T}}_{\lambda,\mu'}^{(F_l, F_k)} D_{\mu'\mu}^\lambda(\alpha, \beta, \gamma) \tag{A2}$$

with Wigner D-functions $D_{\mu\mu'}^\lambda(\alpha, \beta, \gamma) = \langle \lambda\mu | \hat{R}(\alpha, \beta, \gamma) | \lambda\mu' \rangle$.

A2.4. *Hermitian adjoint*: $\left( \hat{\mathbb{T}}_{\lambda\mu}^{(F_l, F_k)} \right)^\dagger = (-1)^{(F_l - F_k + \mu)} \hat{\mathbb{T}}_{\lambda,-\mu}^{(F_k, F_l)}$.

#### A2.1 Proof of A2.1

It is obvious that operators defined over different blocks are orthogonal. Next we check the orthogonality between operators within the same block $(F_l, F_k)$. For simplicity, the superscripts are omitted.

$$\mathrm{Tr}(\hat{\mathbb{T}}_{\lambda\mu} \hat{\mathbb{T}}_{\lambda'\mu'}^\dagger) = \frac{\sqrt{(2\lambda+1)(2\lambda'+1)}}{2F_l+1} \sum_{m_l, m_k} \langle F_k m_k \lambda\mu | F_l m_l \rangle \langle F_k m_k \lambda'\mu' | F_l m_l \rangle, \tag{A3}$$

giving

$$\langle F_k m_k \lambda\mu | F_l m_l \rangle = (-1)^{F_k - m_k} \sqrt{\frac{2F_l+1}{2\lambda+1}} \langle F_l m_l F_k(-m_k) | \lambda\mu \rangle. \tag{A4}$$

The sum can be estimated as

$$\sum_{m_k, m_l} \langle F_k m_k \lambda\mu | F_l m_l \rangle \langle F_k m_k \lambda'\mu' | F_l m_l \rangle = \frac{2F_l+1}{2\lambda+1} \sum_{m_k, m_l} \langle \lambda\mu | F_l m_l F_k m_k \rangle \langle F_l m_l F_k m_k | \lambda'\mu' \rangle = \frac{2F_l+1}{2\lambda+1} \delta_{\lambda\lambda'} \delta_{\mu\mu'}. \tag{A5}$$



### A2.2 Proof of A2.2

Since the operators are orthogonal to each other, they must be linearly independent. The total number of operators over the block $(F_\mathrm{l}, F_\mathrm{k})$ (assuming $F_\mathrm{l} > F_\mathrm{k}$ without lose of generality) is

$$\sum_{\lambda=F_\mathrm{l}-F_\mathrm{k}}^{F_\mathrm{l}+F_\mathrm{k}} 2\lambda + 1 = (2F_\mathrm{l}+1)(2F_\mathrm{k}+1). \tag{A6}$$

Since the number of independent basis operators equals to the number of degrees of freedom over the block (see A1), the defined operator basis is complete.

### A2.3 Proof of A2.3

It is equivalent to prove the following conditions (the superscript $(F_\mathrm{l}, F_\mathrm{k})$ is omitted):

$$[\hat{F}_z, \hat{\mathbb{T}}_{\lambda\mu}] = \mu\hat{\mathbb{T}}_{\lambda\mu},$$

$$[\hat{F}_\pm, \hat{\mathbb{T}}_{\lambda\mu}] = \sqrt{\lambda(\lambda+1)-\mu(\mu\pm1)}\,\hat{\mathbb{T}}_{\lambda,\mu\pm1}, \tag{A7}$$

Proof of the first equation is obvious. For the second part, we have

$$
\begin{aligned}
[\hat{F}_\pm, \hat{\mathbb{T}}_{\lambda\mu}] &= \sum_{m_\mathrm{l}, m_\mathrm{k}} \langle F_\mathrm{k} m_\mathrm{k} \lambda\mu | F_\mathrm{l} m_\mathrm{l}\rangle \left( \hat{F}_\pm | F_\mathrm{l}, m_\mathrm{l}\rangle\langle F_\mathrm{k}, m_\mathrm{k} | - | F_\mathrm{l}, m_\mathrm{l}\rangle\langle F_\mathrm{k}, m_\mathrm{k} | \hat{F}_\pm \right) \\
&= (\sqrt{F_\mathrm{l}(F_\mathrm{l}+1)-m_\mathrm{l}(m_\mathrm{l}\mp1)}\langle F_\mathrm{k} m_\mathrm{k}\lambda\mu | F_\mathrm{l}(m_\mathrm{l}\mp1)\rangle \\
&\quad - \sqrt{F_\mathrm{k}(F_\mathrm{k}+1)-m_\mathrm{k}(m_\mathrm{k}\pm1)}\langle F_\mathrm{k}(m_\mathrm{k}\pm1)\lambda\mu | F_\mathrm{l} m_\mathrm{l}\rangle) | F_\mathrm{l}, m_\mathrm{l}\rangle\langle F_\mathrm{k}, m_\mathrm{k} |.
\end{aligned}
\tag{A8}
$$

We can apply Eq.(A4) and the recursion relationships between Clebsch–Gordan coefficients to find that

$$C_\pm(F_\mathrm{l}, m_\mathrm{l}\mp1)\langle F_\mathrm{l} m_\mathrm{l} F_\mathrm{k}(-m_\mathrm{k}\mp1) | \lambda\mu\rangle + C_\pm(F_\mathrm{k}, -m_\mathrm{k}\mp1)\langle F_\mathrm{l}(m_\mathrm{l}\mp1) F_\mathrm{k}(-m_\mathrm{k}) | \lambda\mu\rangle = C_\pm(\lambda,\mu)\langle F_\mathrm{l} m_\mathrm{l} F_\mathrm{k}(-m_\mathrm{k}) | \lambda\mu\pm1\rangle. \tag{A9}$$

With $C_\pm(F, m) = \sqrt{F(F+1)-m(m\pm1)}$, Eq.(A8) can be simplified to yield the second part of Eq.(A7).

### A2.4 Proof of A2.4

Lastly, we check the Hermitian adjoint property of the basis operators:

$$
\begin{aligned}
\left(\hat{\mathbb{T}}_{\lambda\mu}^{(F_\mathrm{l}, F_\mathrm{k})}\right)^\dagger &= \sqrt{\frac{2\lambda+1}{2F_\mathrm{l}+1}} \sum_{m_\mathrm{l}, m_\mathrm{k}} \langle F_\mathrm{k} m_\mathrm{k}\lambda\mu | F_\mathrm{l} m_\mathrm{l}\rangle | F_\mathrm{k}, m_\mathrm{k}\rangle\langle F_\mathrm{l}, m_\mathrm{l} | \\
\hat{\mathbb{T}}_{\lambda,-\mu}^{(F_\mathrm{k}, F_\mathrm{l})} &= \sqrt{\frac{2\lambda+1}{2F_\mathrm{k}+1}} \sum_{m_\mathrm{l}, m_\mathrm{k}} \langle F_\mathrm{l} m_\mathrm{l}\lambda(-\mu) | F_\mathrm{k} m_\mathrm{k}\rangle | F_\mathrm{k}, m_\mathrm{k}\rangle\langle F_\mathrm{l}, m_\mathrm{l} |,
\end{aligned}
\tag{A10}
$$

Given Eq.(A4) and the symmetry of Clebsch-Gordan coefficients

$$\langle F_\mathrm{l} m_\mathrm{l} F_\mathrm{k}(-m_\mathrm{k}) | \lambda\mu\rangle = \langle F_\mathrm{k} m_\mathrm{k} F_\mathrm{l}(-m_\mathrm{l}) | \lambda(-\mu)\rangle, \tag{A11}$$





we have

$$\left(\hat{\mathbb{T}}_{\lambda\mu}^{(F_1,F_k)}\right)^{\dagger} = (-1)^{(F_1-F_k+\mu)}\hat{\mathbb{T}}_{\lambda,-\mu}^{(F_k,F_1)}. \tag{A12}$$

## Appendix B: Expansion of the density operator over spherical tensor operator basis

The density operator could be expanded over the spherical tensor operator basis:

$$\hat{\rho} = \sum_{F_1,F_k} \sum_{\lambda=|F_1-F_k|}^{F_1+F_k} \sum_{\mu=-\lambda}^{\lambda} \rho_{\lambda\mu}^{(F_1,F_k)}\hat{\mathbb{T}}_{\lambda\mu}^{(F_1,F_k)}. \tag{B1}$$

Applying the orthogonality condition of the basis operators (A2), we have the so-called polarization moments (Auzinsh et al., 2010):

$$\rho_{\lambda\mu}^{(F_1,F_k)} = \mathrm{Tr}\left[\hat{\rho}\left(\hat{\mathbb{T}}_{\lambda\mu}^{(F_1,F_k)}\right)^{\dagger}\right]. \tag{B2}$$

## Appendix C: Completeness of the visualization

The visualization is complete in a sense that there is a one-to-one correspondence between AMP/AMC surfaces and the related density operators. First, as discussed in the main text, the AMP/AMC surface can be uniquely determined from the given density operator. Here we show that the visualized density operator can be reconstructed from the AMP/AMC surface. From Eq. 1 and Eq. 3, the AMP/AMC surfaces can be expressed using the function $f_{m_F}^{(F_1,F_k)}(\theta,\phi) = \mathrm{Tr}\left(\hat{\rho}|F_k,m_F\rangle_{\hat{\mathbf{n}}\,\hat{\mathbf{n}}}\langle F_1,m_F|\right)$ as

$$r^{(F_k,F_k)}(\theta,\phi) = f_{F_k}^{(F_k,F_k)}(\theta,\phi),$$
$$r^{(F_1,F_k)}(\theta,\phi) = e^{i\varphi}f_{m_F}^{(F_1,F_k)}(\theta,\phi) + e^{-i\varphi}f_{m_F}^{(F_k,F_1)}(\theta,\phi). \tag{C1}$$

There are two useful proprieties of the defined function $f_{F_k}^{(F_k,F_k)}$:

C.1. *Expansion over spherical harmonics*

$$f_{m_F}^{(F_1,F_k)} = (-1)^{F_k-m_F}\langle F_1 m_F F_k(-m_F)|\lambda 0\rangle \sum_{\lambda,\mu}\sqrt{\frac{4\pi}{2\lambda+1}}\rho_{\lambda\mu}^{(F_1,F_k)}Y_{\lambda\mu}(\theta,\phi) \tag{C2}$$

where $\rho_{\lambda\mu}^{(F_1,F_k)}$ are the polarization moments (Auzinsh et al., 2010) and $Y_{\lambda\mu}(\theta,\phi)$ are spherical harmonics.

C.2. *Complex conjugation*

$$f_{m_F}^{(F_k,F_1)}(\theta,\phi) = \left(f_{m_F}^{(F_1,F_k)}(\theta,\phi)\right)^{*} \tag{C3}$$

These proprieties (see the proof below) allow extracting the polarization moments of the density operator from the surfaces by decomposing them over spherical harmonics.





## C1  Proof of C.1

The state with a projection $m_F$ along the measurement direction $\hat{\mathbf{n}}$ can be obtained by applying the rotation operator to the
state $|F_k m_F\rangle$ (defined with the projection $m_F$ along the $\hat{\mathbf{z}}$-direction):

$$|F_k m_F\rangle_{\hat{\mathbf{n}}} = \hat{R}(\phi,\theta,0)|F_k m_F\rangle, \tag{C4}$$

where the operator $\hat{R}(\alpha,\beta,\gamma) = e^{-i\alpha \hat{F}_z} e^{-i\beta \hat{F}_y} e^{-i\gamma \hat{F}_z}$ is the quantum mechanical rotation operator written using the $z-y-z$ convention with the Euler angles $\alpha$, $\beta$, and $\gamma$, respectively. Substituting the expansion of the density operator Eq. (B1) into the definition of the function $f_{m_F}^{(F_l, F_k)}(\theta,\phi)$ gives

$$
\begin{aligned}
f_{m_F}^{(F_l, F_k)}(\theta,\phi) &= \sum_{\lambda,\mu} \rho_{\lambda\mu}^{(F_l, F_k)} {}_{\hat{\mathbf{n}}}\langle F_l, m_F | \hat{\mathbb{T}}_{\lambda\mu}^{(F_l, F_k)} | F_k, m_F \rangle_{\hat{\mathbf{n}}} \\
&= \sum_{\lambda,\mu} \rho_{\lambda\mu}^{(F_l, F_k)} \sum_{\mu'} \langle F_l, m_F | \hat{\mathbb{T}}_{\lambda\mu'}^{(F_l, F_k)} | F_k, m_F \rangle D_{\mu'\mu}^{\lambda}(0,-\theta,-\phi),
\end{aligned}
\tag{C5}
$$

where we have employed the sphericalness of the basis operators (A2). The term $\langle F_l, m_F | \hat{\mathbb{T}}_{\lambda\mu'}^{(F_l, F_k)} | F_k, m_F \rangle$ can be easily estimated from the definition A1. Notice that $\mu'$ can only take a zero value. In this particular case, Wigner-D functions are related to spherical harmonics via

$$D_{\mu 0}^{\lambda}(0,-\theta,-\phi) = \left(D_{\mu 0}^{\lambda}(\phi,\theta,0)\right)^* = \sqrt{\frac{4\pi}{2\lambda+1}} Y_{\lambda\mu}(\theta,\phi). \tag{C6}$$

Substituting Eq. (C6) into Eq. (C5) proves the property C.1.

## C2  Proof of C.2

Starting from Eq. (C2), by exchanging $F_k$ and $F_l$ one can find that

$$f_{m_F}^{(F_k, F_l)} = (-1)^{F_l - m_F} \langle F_k m_F F_l (-m_F)|\lambda 0\rangle \sum_{\lambda,\mu} \sqrt{\frac{4\pi}{2\lambda+1}} \rho_{\lambda\mu}^{(F_k, F_l)} Y_{\lambda\mu}(\theta,\phi). \tag{C7}$$

To find out the relationship between $\rho_{\lambda\mu}^{(F_k, F_l)}$ and $\rho_{\lambda,\mu}^{(F_l, F_k)}$, we employ Eq. (A12):

$$
\begin{aligned}
\rho_{\lambda\mu}^{(F_k, F_l)} &= \mathrm{Tr}\left(\hat{\rho}\left(\hat{\mathbb{T}}_{\lambda\mu}^{(F_k, F_l)}\right)^\dagger\right) = (-1)^{(F_k - F_l + \mu)} \mathrm{Tr}\left(\hat{\rho}\,\hat{\mathbb{T}}_{\lambda,-\mu}^{(F_l, F_k)}\right) \\
&= (-1)^{(F_k - F_l + \mu)} \left(\mathrm{Tr}\left(\hat{\rho}\left(\hat{\mathbb{T}}_{\lambda,-\mu}^{(F_l, F_k)}\right)^\dagger\right)\right)^* = (-1)^{(F_k - F_l + \mu)} \left(\rho_{\lambda,-\mu}^{(F_l, F_k)}\right)^*
\end{aligned}
\tag{C8}
$$

By inserting

$$\left(Y_{\lambda\mu}(\theta,\phi)\right)^* = (-1)^\mu\, Y_{\lambda,-\mu}(\theta,\phi) \tag{C9}$$

and

$$\langle F_k m_F F_l (-m_F)|\lambda 0\rangle = \langle F_l m_F F_k (-m_F)|\lambda 0\rangle, \tag{C10}$$

into Eq. (C7) and substituting the summation label $\mu$ by $-\mu$, Eq. (C3) can be proven.





**Appendix D: Rotational symmetry**

Certain rotational symmetries of the density operator can be directly reflected from the visualized AMP/AMC surfaces.

D.1. *Global rotation* Any global rotation can be directly reflected by the rotation of the AMP/AMC surface.

D.2. *Higher-order symmetries* All the visualized AMP/AMC surfaces have a q-fold symmetry along the $\hat{\mathbf{z}}$ axis, then all the
elements of $\hat{\rho}_{m_1,m_k}^{(F_1,F_k)}$ are to be zero except those for which $m_l - m_k = qN$ where $N$ is an integer.

**D1    Proof of D.1**

Assume a global rotation is applied to the density operator, the resulting new density operator can be formulated as

$$\hat{\rho}' = \hat{R}(\alpha,\beta,\gamma)\,\hat{\rho}\,\hat{R}^{\dagger}(\alpha,\beta,\gamma). \tag{D1}$$

Following from the sphericalness of the basis operators, the polarization moments of the rotated density operator are

$$\left(\rho_{\lambda\mu}^{(F_1,F_k)}\right)' = D_{\mu\mu'}^{\lambda}(\alpha,\beta,\gamma)\,\rho_{\lambda\mu'}^{(F_1,F_k)}. \tag{D2}$$

According to Eq. (C2), the function $\left(f_{m_F}^{(F_1,F_k)}\right)'$ which is defined with the new polarization moments can be written as

$$\left(f_{m_F}^{(F_1,F_k)}\right)' = (-1)^{F_k - m_F}\langle F_1 m_F F_k(-m_F)|\lambda 0\rangle \sum_{\lambda,\mu}\sqrt{\frac{4\pi}{2\lambda+1}}\left(\rho_{\lambda\mu}^{(F_1,F_k)}\right)' Y_{\lambda\mu}(\theta,\phi). \tag{D3}$$

On the other hand, the function $\left(f_{m_F}^{(F_1,F_k)}\right)''$ which is rotated from the original function $\left(f_{m_F}^{(F_1,F_k)}\right)$ through the rotating operator is defined as

$$\left(f_{m_F}^{(F_1,F_k)}\right)'' = \hat{R}(\alpha,\beta,\gamma)f_{m_F}^{(F_1,F_k)}. \tag{D4}$$

Following from the rotational symmetry of spherical harmonics,

$$\hat{R}(\alpha,\beta,\gamma)Y_{\lambda\mu}(\theta,\phi) = Y_{\lambda\mu}\left(\hat{R}^{-1}(\alpha,\beta,\gamma)(\theta,\phi)\right) = \sum_{\mu'}Y_{\lambda\mu'}(\theta,\phi)\,D_{\mu'\mu}^{\lambda}(\alpha,\beta,\gamma), \tag{D5}$$

it easy to see that $\left(f_{m_F}^{(F_1,F_k)}\right)'' = \left(f_{m_F}^{(F_1,F_k)}\right)'$. Following from Eq. (C1), the AMP/AMC surfaces obtained through the rotated density operator are exactly the same as those rotated from the original AMP/AMC surfaces.

**D2    Proof of D.2**

If a $q$-fold symmetry is presented for all the AMP/AMC surfaces, the density operator would also have a $q$-fold symmetry since they rotate in sync, i.e.,

$$\hat{\rho}_{m_1,m_k}^{(F_1,F_k)}\,e^{-i(m_1-m_k)\frac{2\pi}{k}} = \hat{\rho}_{m_1,m_k}^{(F_1,F_k)}. \tag{D6}$$

This requires all the elements of $\hat{\rho}_{m_1,m_k}^{(F_1,F_k)}$ to be zero except those for which $m_l - m_k = kN$ where $N$ is an integer.



*Author contributions.*  DAB: conceptualization, JX: analysis, code design, and simulations. JX, DB, and DAB: preparation of the manuscript.

*Competing interests.*  Authors declare no competing interests.

*Acknowledgements.*  This research was supported by Deutsche Forschungsgemeinschaft (DFG), project number 465084791. DAB acknowledges support from the Alexander von Humboldt foundation in the framework of the Sofja Kovalevskaja Award.



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
