# Peer review of "Visualization of Dynamics in Coupled Multi-Spin Systems"

_Magnetic Resonance, 2022_

## Community Comment (CC1)

Dear Jingyan, Dmitry and Danila

Thanks for advancing the MR field! I read your preprint and have a few comments and questions regarding your work.

in the intro you write, "Most NMR textbooks visualize the motion of a single spin-1/2 (or an ensemble of spins) using the Bloch vector."

*I understand this because it gives us a simple rule to explain and describe spin evolution. Moreover, there are modified rules for the week coupling case that use product operator formalism. How do we know what they are useful? Because they give us an excellent tool to predict an effect on paper before measuring it.*

In conclusion, you write: "AMP/AMC surfaces conveniently represent symmetries of density matrices and allow spotting their presence (orientation, alignment, etc.) or absence even when direct analysis of density matrices is not obvious. <..> The presented AMP/AMC surface approach allows visualizing complex dynamics in multispin systems and may find applications for describing hyperpolarization experiments"

*I miss new insights into spin evolution. I do not understand how this approach can help to explain spin evolution simpler or predict something new.*

**Some other comments:**

*Comment 1. I can guess Figure 1c, need corrections. Something like $Fk,mF \rightarrow Fk,mj$, $Fl,mF \rightarrow Fl,mi$, for $mi,mj<Fl$, $rhoFlFk\_mF,mF \rightarrow rhoFlFk\_mi,mj$ and then in the caption only $mF=mi=mj$ were used for visualization.*

*Comment 2. L 135 "Lastly, Fig. 5E, which measures the out-of-phase coherence reveals extra information not covered by Fig. 5B, i.e., the singlet-triplet coherence, $\hat{I}1y\ \hat{I}2x - \hat{I}1x\ \hat{\ } 135\ I2y$, is transiently formed during the experiment."*

*- It was your free choice not to plot o-o-p coherence on B, which you can add, and then again no new information I can gain from the visualization. Or I'm I missing something?*

*Comment 3. Fig 6. T1 and T2 are the unfortunate choices because they have some specific mining in NMR. Maybe small tau instead?*

*Comment 4. Line 150: "Dynamics of the AMC surfaces shown in Fig 6D-F is more intricate. Their motion is a superimposed oscillation of shapes with a period of $T2 = 1/J$ and a slow precession about the x-axis with the above-mentioned period of T1. From the visualization, one may conclude that in-phase and out-of-phase zero quantum coherences give rise to the doublet shown in Fig 6B"*

*- What is your observable? If something evolves with the same frequency it does not mean that it is the reason for the signal (i'm not saying that it is the case here).*

*- You need to explain how the evolution of ZQC results in the observation of the ULF spectrum.*

*Comment 5 : "Code availability. The software code for the graphics shown in this paper is available from the authors upon request"*

*I hope this statement will be not acceptable for any journals and MR will have a strict policy on publishing all scripts together with the paper or EU-repositories with doi like zenodo.org*

*With kind regards*

*Andrey Pravdivtsev*

---

## Editor Comment (EC2)

Further on my comment on MR-2022-9:

Figure 7 from 'Cross-Correlation of Chemical Shift Anisotropy and Dipolar Interactions in Methyl Protons Investigated by Selective Nuclear Magnetic Resonance Spectroscopy' by N. Müller and G. Bodenhausen, J. Chem. Phys. 98, 6062-6069 (1993). This may serve as a warning for future generations against over-sophistication of graphical conventions.

[Figure]

FIG. 7. Three-dimensional coherence transfer diagram for the pulse sequence in Fig. 6 applied to an $AM_3$ system. The time axis runs from front to back. Horizontal levels represent tensor ranks $l$, while vertical planes represent coherence orders $p$. The upper wedge-shaped part of the diagram represents states $T_{lp}^M$ that are accessible to the three equivalent methyl protons; the lower part corresponds to the states $T_{lp}^A$ accesible to the $A$ proton. RF pulses (indicated by circled numbers corresponding to Fig. 6) affect changes between vertical planes, while spin–spin coupling and multiexponential relaxation processes may be described by moving between different horizontal levels.

---

## Author Response (AR1)

We thank the referees for their thoughtful comments and suggestions. Considering these comments, we made the following modifications:

1. The abstract has been updated;

2. The introduction has been updated;

3. The results section has been updated;

4. A section (Section 3.1) discussing the symmetry and equivalence between the density operator and the plotted surfaces was added;

5. Discussions about how the visualization help us in understanding the ZULF *J*-spectra was added to section 3.2;

6. A section (Section 3.2) comparing our generalized measurement-based visualization and DROPS approach was added;

7. A step-by step instruction was added to appendix F.

Please see below the detailed response to the referees' questions.

**Referee #1**

The authors provide a new way of visualizing density matrix components during pulse sequences in multispin system using angular momentum probability surfaces. The approach appears to be superior to the previously described DROPS method. One component of the approach involves coupling angular momenta, but it seems that the actual visualization is not very intuitive to follow. Either way, it seems the authors provide some useful examples. I am wondering whether the authors could identify cases where this approach can produce some level of intuition that would exceed the level of intuition one would get from examining the spin components directly.

We thank the referee for assessing our approach as superior to the previously described methods. One particular case where the presented approach may produce an enhanced level of intuition is the presented near-zero-field NMR experiment (Figure 5). Additional discussion is added to the revised version of the manuscript. Initial magnetization of two coupled spins (with gyromagnetic ratios g1 and g2) in high magnetic field is (g1\*I1z + g2\*I2z). This state can be presented as a sum of symmetric and antisymmetric components: 0.5\*(g1+g2)\*(I1z+I2z) and 0.5\*(g1-g2)\*(I1z-I2z). Remarkably, both of these components can be detected by a magnetometer in ZULF NMR experiments. The first symmetric component corresponds to the orientation of a collective magnetic moment (I1z+I2z) and its visualization is related to an AMP surface presented by the Fig.6C (however, the surface represents probability and not exactly the measured property). This component typically corresponds to static magnetization at zero field. The experiment shown in Fig. 5C makes this component precessing with Larmor frequency 0.5\*(g1+g2)\*Bx upon application of the magnetic field Bx. The second component of decomposition represents a zero-quantum coherence (I1z-I2z) which is directly detectable by the magnetometer. Intersection of the plotted surface with any axis is a direct representation of a measured signal by a magnetometer along that axis.

Changes made to the manuscript: None.

**Referee #2**

The paper describes a visualization tool for the representation of density operators in multiple-spin systems. The approach is inspired by the DROPS software of Glaser and co-workers, and shares with that work most strengths and weaknesses. The strength is a graphical representation which might possibly lead to a helpful visualization of complex spin dynamics, sidestepping the need for complex mathematics, and possibly help inspire new procedures or give new insights. The main weakness is that although the tool allows a pretty graphical representation of the mathematics, it does not replace the mathematics, at least not as far as I can see. So the result seems to be pretty graphics (which I am all for) but not with evident real utility, in contrast to the Bloch vector picture, from which many NMR effects and experiments have been derived. So I am not yet convinced of the utility of this representation.

Authors' reply: We thank the referee for the critique. We note that the generalized measurement-based visualization approach provide us with valuable information which is not easy to grasp through direct observation of numerical values of the density-matrix elements. First, the action of global rotations applied to the density operator is directly reflected by the rotations of the plotted surfaces (see Fig. 3). Second, there is a close relationship between the density matrix coherence and the symmetries of the plotted surfaces (see Fig. 4). Third, our visualization allows one to quantitively understand the measured ZULF NMR spectrum by looking at the intersection of the surface with an axis representing sensitive direction of a magnetometer. As an example, the ZULF *J*-spectrum of an AX system is now pictorially explained through our visualization. Specifically, we explained the small asymmetry of the doublet centered at *J* which is not easy to grasp through the product operator formalism.

Changes made to the manuscript: The Section 3.1 was added to discuss in detail the rotational and symmetry properties of the visualized surfaces. Additionally, the ZULF section (3.2) was updated to discuss direct applicability of the plotted surface for assessing the measured ZULF NMR signal.

Furthermore I cannot see exactly how it works, and the authors do not help since they choose a mathematically dense exposition which is very hard to follow, right from the beginning. Despite my best efforts I cannot understand equation 1 and the following equations. It may be that the terms used by the authors are self-evident to the atomic physics community but I suspect that most readers of this journal will, like myself, struggle greatly with it. For this to work the authors must make far greater efforts to express themselves in language comprehensible to us mere magnetic resonance mortals. What on earth is the "block (Fk, Fk)"? Scientists on the same level of mathematical physics as myself will need to be led far more slowly through this material, using helpful simple examples on the way.

Authors' reply: We thank the referee for the feedback regarding our work. We significantly improved the text by simplifying some mathematical notations and by using terminology which is more familiar to the NMR community. In simple words, our visualization is now performed by plotting measurements with zeroquantum Hermitian operators rotated along various directions.

Changes made to the manuscripts: Introduction, Results, and Discussion sections were modified in multiple places to incorporate better explanation of terminology and to include simple examples using spin-1/2 pairs. The Appendix F now includes step-by-step implementation of the visualization making it easier to understand.

In addition the authors introduce the term "angular momentum coherence (AMC)". I suspect that the term coherence is used here to mean something very different from its standard usage in magnetic resonance, as defined by Ernst and co-workers (an off-diagonal density operator term, when written in the Hamiltonian eigenbasis). I am not sure though since I cannot follow the authors' meaning. In general I do not think a redefinition or loose usage of this fundamental term is advisable.

Authors' reply: We thank the referee for the critique. Despite the visualized coherences were indeed off-diagonal elements of the density operator written in the total angular momentum basis (i.e., Hamiltonian eigenbasis at zero magnetic field), we agree that excessive use of the "coherence" terminology may not add additional value to the paper. For this reason, we abandon the term AMC (angular momentum coherence) surfaces in the updated version of the paper and refer to the visualized surfaces directly via the measurement operators.

Changes made to the manuscript: Multiple changes made to the text to avoid AMC terminology.

In summary I am sympathetic to the aims of this paper but find the presentation impossible to follow. In addition, I am far from convinced of its usefulness, but recognise that it could be of value if explained well enough and made sufficiently accessible.

Authors' reply: we thank the referee; the text was significantly improved. Changes made to the manuscript: multiple changes of the text, see above.

On the issue of accessibility, I agree with another referee that it is no longer acceptable, for work of this kind, to say that the code is available on request.

Authors' reply: We thank the referee for the comment. However, the statement "The software code for the graphics shown in this paper is available from the authors on reasonable request" was a direct copy (except deletion of the word reasonable) from the following article published in *Magn. Reson.* (see https://mr.copernicus.org/articles/2/395/2021/). Nonetheless, we gladly provide the code in the revised version of the manuscript.

Changes made to the manuscript: the text was updated to "The software code for the graphics shown in this paper is available in Supporting Information."

A further comment: The article emphasizes the total spin angular momentum quantum number (denoted F, I believe). Off-diagonal density operator elements spanning states with different values of F appear to be called "angular momentum coherences" (AMCs). This nomenclature and analysis might be appropriate for atomic physics, where the Hamiltonian has isotropic, or nearisotropic symmetry. However this situation is rarely encountered in magnetic resonance of bulk matter, since we hardly ever deal with isotropic systems. Trivially, the application of a strong magnetic field breaks isotropic symmetry (leading, amongst other things, to the Zeeman effect, upon which most magnetic resonance is based). Even the solution NMR of isotropic liquids does not involve an isotropic Hamiltonian. Very often, chemical shift differences and other interactions break the symmetry further. In most cases these are not small perturbations but conpletely break the isotropy of the spin Hamiltonian. There are rare exceptions, such as zero-field NMR. Since high-field NMR almost always uses Hamiltonian eigenstates that do not have well-defined values of F, I do wonder what utility this diagrammatic approach might have. Furthermore, although the concept of AMC's "angular momentum coherences" might possibly mean something in atomic physics, I suspect that it has no, or little, relevance to the vast majority of magnetic resonance experiments, and probably conflicts with the conventional use of the term coherence in magnetic resonance - namely a coherent superposition of Hamiltonian eigenstates.

Authors' reply: We thank the referee for the critique. We removed terminology of the AMC surfaces and now clearly point out that the method is of utility when total angular momentum basis is a convenient basis for describing NMR experiments. All three examples from the paper fall into that category.

Changes made to the manuscript: the abstract and the Results section were updated to emphasize that the presented approach "finds particular utility when the total angular momentum basis is used for describing the Hamiltonian".

I think the article will not be appropriate for the magnetic resonance community unless these sharp differences between the atomic spectroscopy and bulk magnetic resonance contexts are highlighted more clearly.

Authors' reply: We disagree with the referee, but we significantly improved the paper to make it more accessible for the magnetic resonance community.

Changes made to the manuscript: the text was modified in multiple places.

**Referee #3**

The stated aim of the manuscript is to extend the concept of angular momentum probability ("AMP") surfaces (which have been shown to be useful in the understanding of atomic physics experiments) by so-called angular momentum coherence ("AMC") surfaces. The authors show that this makes it possible to create three-dimensional graphical representations of the density operator of coupled spins, which is illustrated for three concrete NMR pulse sequences. The authors also show that the suggested visualization is complete in the sense that the density operator can be reconstructed from a full set of "AMP" and "AMC" surfaces. The mathematical basis of the visualization approach appears to be solid (but I agree with the comments of other reviewers that the presentation of the derivations and proofs should probably be adapted to the readers of Magnetic Resonance).

Authors' reply: We thank the referee for assessing mathematical basis of our work as solid.

**Changes made to the manuscript: None.**

In my view, the most important weak point of the current manuscript is a thorough discussion of how the presented approach is related to similar visualization techniques that have been introduced before for the visualization of coupled spin/angular momentum dynamics. In particular, the readers (as well as the referees and the editor) will be interested to see what are truly novel aspects in terms of the visualization approach or novel applications and to give a proper account of closely related previous work. (Before I discuss these aspects in more detail below, I would like to point out that even if the visualization should be closely related (or even be essentially identical) to previously published approaches, I would still be in favor to publish a revised

manuscript in which these points are considered, because as far as I see, the proposed visualization variant has not been applied to concrete NMR settings yet and it should be interesting and useful for the readers to see whether or not it could have advantages compared to other visualizations approaches.)

**Point #1:**

Before addressing the main point (the potential novelty of the "AMC" surface representation). I would like to briefly discuss the established "AMP" surface representation for uncoupled spins or atoms or molecules with arbitrary angular momentum (called F in the nomenclature used in the manuscript or J in other settings). I think the proper context in which this work should be placed is the general field of phase space representations, at least by referring to the books by W. P. Schleich, Quantum Optics in Phase Space(Wiley, New York, 2001), C. K. Zachos, D. B. Fairlie, and T. L. Curtright, Quantum Mechanics in Phase Space: An Overview with Selected Papers(World Scientific, Singapore, 2005). F. E. Schroeck, Jr., Quantum Mechanics on Phase Space(Springer, New York, 2013). T. L. Curtright, D. B. Fairlie, and C. K. Zachos, A Con- cise Treatise on Quantum Mechanics in Phase Space(World Scientific, Singapore, 2014 and the recent review by R. P. Rundle and M. J. Everitt in Adv. Quantum Technol. 2100016 (2021). For a general discussion and comparison of different families of phase-space representations, I refer in particular to the recent paper (B. Koczor, R. Zeier, S. J. Glaser, Continuous Phase-Space Representations for Finite-Dimensional Quantum States and their Tomography", Phys. Rev. A 101, 022318, 2020) and references therein.

In (Koczor et al., 2020), the focus is the family of so-called s-parametrized phase-space functions of which the widely-used Glauber P function (with s=1), the Wigner W function (with s=0) and the Husimi Q function (with s=-1) are special cases. The paper gives an overview how the plethora of different finite-and infinite-dimensional phase space representations are related and can be mapped to each other. In particular, see Fig. 2 of (Koczor et al., 2020) for a graphical overview, section III on phase-space representations for spins and Ref. 76 (Stratonovich, 1956), Ref. 58 (Argawal, 1981), Ref. 77 (Varilly et al., 1989), and Ref. 57 (Brif et al, 1999).

As far as I can see (but please correct me if I am wrong), the definition of the "AMP" representation appears to be identical with the finite-dimensional version of the Husimi Q function, which to my knowledge was first defined (under different names) in the early 1980s (Argawal, 1981). Surprisingly, the close connection (if not identity) of the "AMP" surfaces and the Husimi Q function for finite-dimensional quantum systems seems to have gone unnoticed – or at least has apparently not been pointed out in the previous literature (and I need to apologize that before reviewing the current manuscript, I was not aware of the

"AMP" representation and therefore we had not explicitly mentioned it in (Koczor et al., 2020).

So far, I have not received through the library the book (M. Auzinsh, D. Budker, S. Rochester, 2010) cited in the manuscript by Xu et al. and could not check if more details are given there on the relation between "AMP" surfaces and other phase-space representations. However, I found Simon Rochester's thesis from 2010 online, in which he points out that Chapters 2-5 of his thesis are largely identical to sections in the book, of which he is an author. In chapter 2 of his thesis (section 2.3.3), it is pointed out that the expansion of "AMP" functions in terms of spherical harmonics in Eq. 2.39 is "quite similar" to the Wigner function for angular momentum states given in Eq. 2.40 (from Dowling et al., 1994). I assume that this is also pointed out in the book (M. Auzinsh, D. Budker, S. Rochester, 2010). In this section of the thesis is also correctly stated that the essential difference between the expansion of the Wigner function and the "AMP" function is "that the contributions of polarization moments of various ranks are weighted differently", resulting in the fact that "AMP" functions are non-negative. However, this is exactly the property of the Husimi Q function, which has the same weighting factors as the "AMP" function and the same physical interpretation, see also (Koczor et al., 2020), where the s-dependent weighting factors are explicitly given for the family of s-parametrized phasespace functions (see Eq. 5 and Fig. 3).

To summarize point #1, I think it would be very beneficial to point out in the paper that the "AMP" functions are in fact identical with the Husimi Q function and to add corresponding original references and Ref. (Koczor et al., 2020), in order to avoid potential confusion that can arise if different names are used for the same concept in closely related scientific communities and to clearly define the relation between Husimi Q/"AMP" functions and other members of the family of s-parametrized phase-space functions. It would also be interesting to state in the manuscript whether or not my impression is true that the definition of (Argawal, 1981) predates the (identical) definition of what is called the "AMP" function.

Authors' reply: We thank the referee for the constructive comment. The manuscript was improved by citing all mentioned literature. In addition, the equivalence between the AMPS and the Husimi Q-function was indeed established and the proof is now presented in the Appendix G.

Changes made to the manuscript: Introduction section was modified to include citation of the highly relevant literature. Appendix G was added to show the proof of the equivalence between Husimi Q-function and AMP surface function.

**Point #2:**

This point concerns the question about the degree of novelty of the "AMC" surface representation. In the introduction, of the manuscript, the authors mention the "DROPS" representation introduced by (Garon et al., 2015), but state that "while the approach conveniently reflects the dynamics and symmetry of individual spins ..., it is challenging to extract information from the drops representing systems of equivalent (or nearly equivalent) spins" and that the approach based on "AMP" and "AMC" surfaces is introduced "to address these limitations". This section raises several issues:

Point 2.1: To avoid any confusion of nomenclature, let me first address a minor point:

In the introduction of the manuscript, it is implied that "spin drops" is equivalent to the "DROPS" representation, which is not the case and the two concepts should be clearly distinguished:

"DROPS" is a general approach to visualize operators of general spin systems (coupled or not) that was introduced in the paper (Garon et al., 2015). In addition, the implementation of the "DROPS" representation and visualization (based both on the "LISA basis and the multipole basis") in a Mathematica package is publicly available by downloading the file "DROPS\_1.0.zip" at https://www.ch.nat.tum.de/en/ocnmr/media-reports/downloads.

"SpinDrops" is the name of an interactive software package that is freely available for the community (see www.spindrops.org), which (in addition to other approaches) also provides the option to visualize the density operator (as well as Hamiltonians, propagators etc.) using a "DROPS" representation.

Authors' reply: We thank the referee for the comment. The manuscript was updated in multiple places to use the correct terminology (DROPS representation).

Changes made to the manuscript: Multiple changes in the Introduction section.

Point 2.2:

The "DROPS" representation is a general mapping between operators and a set of spherical functions (so-called droplets). As pointed out in (Garon et al., 2015), it is a Wigner-type (generalized) phase-space representations, which is applicable for arbitrary spin systems. In (Garon et al., 2015), the detailed mapping (which is based on symmetry-adapted spherical tensors) was explicitly presented for systems consisting of up to three coupled spins ½. More

recently, explicit symmetry-adapted spherical tensors were constructed based on which systems consisting of up to six spins ½ as well as for coupled spins > ½, based on which arbitrary operators in such systems can be represented and visualized (see Leiner, Zeier, Glaser, "Symmetry-Adapted Decomposition of Tensor Operators and the Visualization of Coupled Spin Systems", J. Phys. A: Math. Theor. 53, 495301, 2020).

Both in (Garon et al., 2015) and in (Leiner et al, 2020), we focused on the "DROPS" representation based on the so-called "LISA" basis of spherical tensor operators (with defined linearity, subsystem, and auxiliary criteria, such as permutation symmetry), which is specifically constructed to visualize the individual spin contributions, which are relevant in most high-field NMR experiments. (In fact, due to the symmetry-adapted "LISA" basis, also magnetically equivalent spins can be efficiently represented).

However, in addition to the "LISA" basis, in section VII and VIII, appendix F, Tables II, VI and VII, Figures 6 and 12 of (Garon et al., 2015), we also explicitly and extensively discussed "DROPS" representations based on so-called multipole spherical tensor basis operators, generalizing a visualization introduced by Merkel et al. in 2008 (Ref. 18 in Garon et al., 2015). I did not have a chance to make an in-depth comparison yet, but I appears that this "DROPS" representation based on the multipole basis is essentially identical (or at least very closely related) to the "AMC" functions. Note that in particular the form of the "multipole operators" corresponding to transitions between blocks with different total angular momentum quantum numbers defined in Eq. (F1) in appendix F of (Garon et al., 2015) appears to be identical to the corresponding definition of the spherical tensor operators in Eq. (A1) in appendix A of the manuscript by Xu, Budker and Barskiy on which the definition of the "AMC" surface functions is based. In fact, both the "multipole operators defined by (Garon et al., 2015) and the spherical tensor operators in Eq. (A1) of the manuscript by Xu et al. differ from the tensor operators in the "LISA" basis in not having a defined particle number (i.e., "linearity") and inducing a different grouping into droplets. One (apparently trivial) difference seems to be that the droplets corresponding to transitions from "F\_I" to "F\_k" and from "F\_k" to "F\_I" are separately displayed in droplets in of (Garon et al., 2018) and (Merkel et al., 2008), whereas they are merged in the manuscript by Xu et al, whereas droplets corresponding to zero-quantum phase phi=0 and phi=pi/2 are displayed separately. Another small difference is that (at least the droplets corresponding to the diagonal blocks in Fig. 1 are non-negative (corresponding to a Husimi Q function representing angular momentum pointing probabilities), whereas in the "DROPS" representation based on multipole operators, the droplets corresponding to the diagonal blocks can have negative values (corresponding to a Wigner function, representing the expectation values of socalled axial tensor operators). However, as discussed above in point #1 (and

as indicated in the thesis of Simon Rochester and also discussed in (Koczor et al., 2020)), it is straight-forward to transform between a Husimi Q and a Wigner W representation by simply changing the rank-dependent weighting factors of the polarization moments.

To summarize point 2: It would be very helpful for the readers to clearly state how closely the combined "AMP" and "AMC" surface representation is related to the "DROPS" representation based on multipole operators and to point out potential differences and whether or not the differences are significant.

Authors' reply: We thank the referee for the thoughtful comment. The manuscript was updated in multiple places. First, the terminology of AMC (angular momentum coherence) surfaces has been abandoned and now the visualization is based on using general zero-quantum Hermitian operators for plotting the surfaces. This makes the surfaces represent measurable properties as now discussed in the section 3.2.

Decomposition of the density matrix into blocks corresponding to the total angular momentum quantum numbers F is the same as in (Garon et al., 2015, when using multipole tensor operator basis), however, the visualization procedure is different. DROPS approach is a Wigner-type representation (visualization is complete but the composed surfaces do not directly represent measurable properties of the density matrix), while our approach is measurement-based representation (visualization is complete and the radius of the surface directly represents a measurable property of the density matrix).

Changes made to the manuscript: Multiple changes in many places throughout the paper.

In particular, the following statements need to be corrected:

Line 30 "... it is challenging to extract information from the drops representing systems of equivalent (or nearly equivalent) spins." This is clearly not the case, in particular if the DROPS representation based on multipole operators is applied.

Authors' reply: we thank the referee for the comment. We rewrote the sentence.

Changes made to the manuscript: The updated sentence reads "While the DROPS approach could be generalized to isotropic systems by using multiple tensor operator basis, it is challenging to extract the information corresponding to measurable properties from the DROPS with complicated colors."

Line 34: "... To address these limitations, we introduce ...". As pointed out

above, the DROPS representation based on multipole operators does not have the stated limitations

Authors' reply: we thank the referee for the comment. We rewrote the sentence. We want to note that DROPS representation may be discriminatory with respect to color-blindness of some groups of the population.

Changes made to the manuscript: The updated sentence reads "While the DROPS approach could be generalized to isotropic systems by using multiple tensor operator basis, it is challenging to extract the information corresponding to measurable properties from the DROPS with complicated colors."

For the presented NMR examples, comparison of the "AMP/AMC" with the standard DROPS representation based on the "LISA" basis and/or with the DROPS representation based on multipole operators is not mandatory, but could be quite useful (at least for one of the presented NMR examples), to make it possible for the readers to judge (potential) advantages or disadvantages of the different visualization approaches.

Authors' reply: we thank the referee for the valuable comment. We performed additional simulations and now present a direct comparison between our visualization approach and the DROPS approach (see Section 3.5) when applied to the ZULF NMR experiment.

Changes made to the manuscript: a new section (3.5) was added per referee's suggestion.